# Vibration Analysis of Pulse-Width-Modulated Nozzles in Vineyard Blast Sprayers

**Coral Ortiz ***[ID]**, Antonio Torregrosa** [ID]**, Verónica Saiz-Rubio** [ID] **and Francisco Rovira-Más** [ID]

Departamento de Ingeniería Rural y Agroalimentaria, Universitat Politècnica de València, Camino de Vera s/n, 46022 Valencia, Spain; torregro@dmta.upv.es (A.T.); vesairu@upv.es (V.S.-R.); frovira@dmta.upv.es (F.R.-M.)
***** Correspondence: cortiz@dmta.upv.es

**Abstract:** Spraying systems to protect crops against pests are still necessary to maintain food production at the rates demanded by the current population. However, today, it is crucial to use precision agriculture to reduce the negative effects of pesticides and other agrochemicals such as fungicides. In particular, pressure fluctuations related to transient states when using pulse-width-modulated nozzles (PMW) have been reported to decrease the accuracy of preset flow rates in air-assisted orchard sprayers. The objective of this paper is to analyze the vibrations induced in the spraying system of a vineyard blast sprayer controlled by pulse-width-modulated nozzles, considering the instantaneous duty cycle (DC) as the control variable. An air-assisted vineyard sprayer was modified to host 24 solenoid shutoff valves with hollow disc–cone nozzles. A triaxial accelerometer was mounted to track the effect of duty cycle (20%, 30%, 50%, and 70%). In addition to accelerations, high-speed images were recorded, and the pressure according to time and the flow were estimated. The hydraulic system of the sprayer, when controlled in real time by the PWM solenoids, created pulsating impacts at the nozzle level with the same frequency of 10 Hz of the PMW system. The impact effect was significantly higher for low duty cycles under 40% DC. In addition, to demonstrate the inaccuracy of opening and closing the valves at a precisely specified time, this study also confirmed the divergence between the theoretical duty cycles commanded by the sprayer's control unit and the actual ones measured in real time. The results of the analysis showed the difficulty of opening and closing the valves with precision to obtain accurate duty cycles in the practical implementation of smart sprayers and the importance of understanding the vibration effects of pulses in arrangements of multiple PWM nozzles working simultaneously.

**Keywords:** PWM nozzle; vibration; solenoid valve; PWM duty cycle; blast sprayer; transient response

## 1. Introduction

For many decades, pesticides have been widely used in agriculture to control natural crop hazards and provide an adequate food supply for people and high-quality foodproducts [1,2].

Chemical protection against pests, pathogens, and weeds is necessary to maintain large-scale crop production in Europe [1]. The use of pesticides is still necessary to provide enough food supply for the world population [2]. In vineyards and orchards, the most efficient method of pulverization is hydropneumatic, which consists of a hydraulic system coupled with an air fan [3]. It has been proven that a considerable amount of the applied phytosanitary product is lost [4–6], and according to previous studies, only 30–40% of the pesticide droplets are deposited on the proposed target [7]. As a result, it is essential to use precision viticulture technology to reduce the long-term impacts of viticultural management practices [8]. Different mitigation strategies for blast spraying vineyards and other orchard crops have been addressed to better adjust the number of products and the localization of the product's application, with the intention of reducing human and environmental hazards. The combination of several mitigation measures and the importance of

involving farmers have been remarked by [9]. Different meteorological and technical factors have proven to affect spray drift [10]. The necessity of controlling adequate deposition in the canopy has been marked by some authors [11,12]. They suggested to regulate the deposition according to the specifications of the treatment and proposed a methodology for drift measurement based on LIDAR sensors. Drift reduction in spray applications with pneumatic sprayers has been addressed through the design of new prototypes matching environmental requirements and treatment specifications [13,14]. Precision agriculture has been presented as a possible solution to apply pesticides to the target area and reduce product usage while maintaining treatment efficacy [15]. Pulse-width-modulated (PWM) solenoid valves were developed to control the amount of product of spray nozzles independently to match tree structures in variable-rate spraying applications [16]. Pressure fluctuations result in inaccurate flow rates for air-assisted orchard sprayers, and previous studies have reported the variability in flow rates due to pressure fluctuations [17,18]. In order to assess the effective control of insects and diseases by a PWM system, ref. [19] tested three spraying systems using an air blast sprayer on two-year old apple trees and demonstrated that PWM assured the density of droplets required to control insects and diseases and also reduced ground and airborne drifts. The consistency and accuracy of the spray applications using PWM solenoid nozzles was studied by [20] using different sensors to record pressures upstream and downstream of the PWM valve, as well as different flow rates and spray angles. With the idea of examining sprayers using PWM nozzles, ref. [21] addressed pressure fluctuations in sprayer applications using PWM nozzles and remarked the importance of considering pressure variations and the time required to reach equilibrium. In a different study, ref. [22] addressed three different PWM nozzle control systems and found significant differences among the three systems in pressure drops, stabilized pressure application time, and flow rate, confirming that PWM systems can deliver incorrect flow rates caused by variations in the control time cycle determined by peak time, stabilized application pressure time, and fall time. Similarly, ref. [21] focused on the effect of nozzle body volume on pressure dynamics when using PWM nozzles at different application volumes. To test this, they developed a static test and registered pressures at 345 kPa and a 50% duty cycle for different nozzle orifice sizes, estimating the damping ratio. They concluded that previous studies may have underestimated the importance of pressure oscillations and the resulting time to reach equilibrium, suggesting the necessity of improving pressure measuring systems. In the same line, ref. [23] studied the effect of pressure on duty cycle for commercial blast sprayers modified with PWM nozzles under laboratory conditions. Four different duty cycles (25%, 50%, 75%, and 100%) and four system pressures (400 kPa, 500 kPa, 600 kPa, and 700 kPa) were combined to monitor the flow rate of several nozzles. The opening and closing of the nozzles for each duty cycle analyzed was recorded with a high-speed video camera. The experiments revealed substantial pressure variations at the nozzle level, which pointed to significant discrepancies between the rated duty cycles and the actual duty cycles applied. Additionally, severe vibrations transmitted to the manometer gauges and the pipes were observed as soon as the PWM nozzles were activated, leading to the conclusion that further research would be needed before confirming the suitability of PWM systems for regulating nozzle flow rates without modifying the system pressure for commercial blast sprayers. Several studies have reported the use of accelerometers to analyze vibrations in agricultural machinery [24–26], in particular the analysis of boom vibrations in spray bars to improve the performance of spraying operations [27–29]. The implementation of pneumatic dampers to reduce vibrations has been reported for multiple agricultural applications, too [30–33]. The objective of this paper is to analyze the effects of vibration in the hydraulic system of a vineyard blast sprayer controlled by pulse-width-modulated nozzles. The analysis focuses on the pulsing impacts induced by PWM solenoids and the accuracy of applying a range of relevant duty cycles.

## 2. Materials and Methods

An air-assisted vineyard sprayer, completely redesigned to allow full automatic control of the spray via an onboard computer reading prescription maps and hydraulically modified to host 24 solenoid shutoff valves (115880 e-ChemSaver, TeeJet Technologies, Glendale Heights, IL, USA) with hollow disc–cone nozzles, was evaluated in static laboratory conditions. The sprayer comprises four foldable arms, each representing an individually controlled sector of six PWM nozzles, labeled $S_1$, $S_2$, $S_3$, and $S_4$, as indicated in Figure 1. During the tests, all the sectors and nozzles were open to mimic real conditions, even though measurements were conducted on specific nozzles according to the experimental design followed. A manual pressure gauge and a digital pressure sensor (Gems Sensors & Controls, Plainville, CT, USA) were included in the hydraulic system of the sprayer. The nozzle flow rates were initially estimated from the manufacturer tables for each registered pressure and verified by weighing the water accumulated in containers individually connected to specific nozzles.

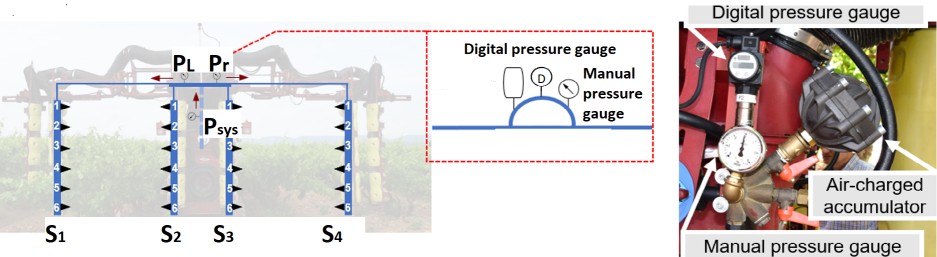

**Figure 1.** Vineyard air-assisted sprayer used for the vibration tests with its four active arms (sectors, $S_1$, $S_2$, $S_3$, and $S_4$) and two measurement units, one per side, each formed by a conventional pressure gauge ($P_{sys}$) and a digital pressure sensor (pressures $P_r$ and $P_l$).

A 3-axis accelerometer with a range of ±500 g, sensitivity 10 mV/g, 3.3 g of weight, and a cube shape of 0.01 m length (Kistler 8763A500, Sindelfingen, Germany) was placed in one of the sprayer sectors, $S_1$, close to the nozzle, as depicted in Figure 2. The vibration plane was perpendicular to the pipe axis (radial axis, $X$ and $Y$), and the non-vibration direction was aligned with the pipe axis ($Z$ axis).

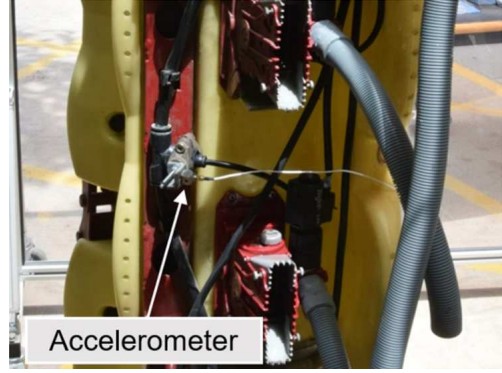

**Figure 2.** Measuring point for $S_1$ accelerometer close to the nozzle.

The signals from the accelerometers, acquired by the oscilloscope, were analyzed using Picoscope Oscilloscope Software 6 (Picotechnology, https://www.picotech.com/products/oscilloscope, accessed on 10 June 2023). The vibration parameters peak to peak and weighted RMS were calculated after applying a low pass filter (100 Hz) and signal averaging. The analysis of the acceleration signal in the frequency domain was performed with a fast Fourier transformation with 401 lines in a frequency range of 0–156.2 Hz. A high-speed digital color video camera (CASIO EX-F1, Tokyo, Japan) recorded the spraying actuation and the oscillations of the manual pressure gauge during the tests at a rate of 300 frames/s. The software Windows Movie Maker (Movie Maker, Seattle, WA, USA) was

used to analyze individual frames. The experimental design studied three commanded duty cycles, 20%, 50%, and 70%. In order to analyze the representativeness of the vibration tested nozzle, a test was carried out to check the homogeneity among nozzles and sectors. Spray flows were measured from four nozzles in high position and four nozzles in low position. In particular, the flow was measured in nozzles 2 and 5 for all four sectors. With all nozzles open, buckets and test tubes were connected to the nozzles with flexible pipes. Flows were estimated by weighing the water sprayed for a given period of time measured with a handheld chronometer. The flow of each nozzle was measured in L·min$^{-1}$, considering a water density of 0.997 kg·L$^{-1}$, the weight of each bucket being 0.51 kg, and the weight of each test tube being 0.47 kg. The overall flow of the system was measured with a flow rate sensor (FT-08NEXWULEE-5, FTI Flow Technology Inc., Tempe, AZ, USA). The first trial required 5.5 min at a DC of 20%, and the second trial took 3 min at a DC of 60%. The vibration parameters peak to peak and weighted RMS were calculated after applying a low pass filter at 100 Hz and performing signal averaging. The time interval between valve opening and valve closing was measured using the accelerometer signals at ten different vibration times per duty cycle (for the three duty cycles). The analysis of individual video frames was carried out with Windows Movie Maker (Windows, Seattle, WA, USA), and the analysis of the variance (ANOVA) for numerical data was performed with Statgraphics Centurion XVIII (Statgraphics, The Plains, VA, USA).

## 3. Results

### 3.1. Homogeneity among Nozzles and Sectors

The range at 20% DC ranged from 0.16 L·min$^{-1}$ to 0.25 L·min$^{-1}$, being the lower values at the extreme sectors $S_1$ and $S_4$, while values over 0.2 L·min$^{-1}$ were found at central sectors $S_2$ and $S_3$ (Figure 3). The range at 60% DC went from 0.40 L·min$^{-1}$ to 0.48 L·min$^{-1}$. Flow was evenly distributed, with smaller differences. Rounding flow values to one decimal, all flow values at 20% DC were 0.2 L/min, while at 60% DC, they were were 0.4 L·min$^{-1}$, except Q5-$S_2$, which was 0.5 L·min$^{-1}$. Despite the slight discrepancies in the flows among sectors, the sprayer can be used for variable rate applications.

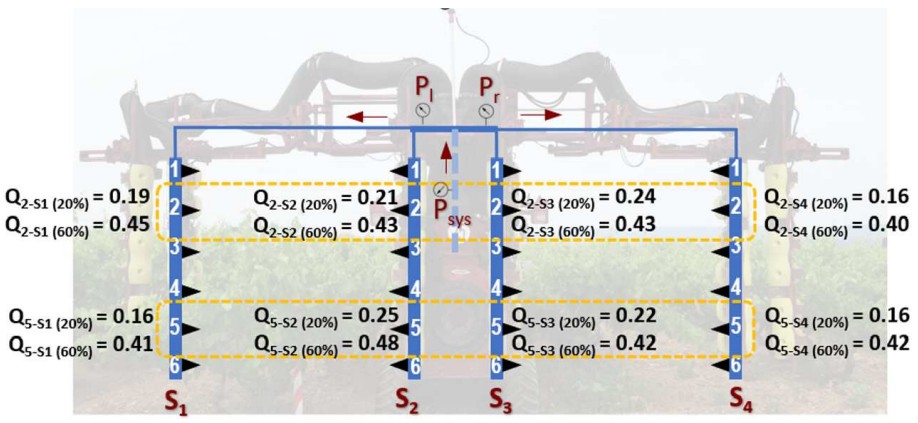

**Figure 3.** Measured flows (L·min$^{-1}$) for the tested nozzles of sectors $S_1$, $S_2$, $S_3$, and $S_4$.

### 3.2. Pressure Oscillations

The video images tracking the oscillations of the conventional pressure gauge helped to analyze the evolution of pressure with time. The pressure gauge images from the high-speed video recordings were analyzed to register the pressure values according to time. The oscillatory behavior of pressure was confirmed by the conventional pressure gauge, as plotted in Figure 4. The registered pressure presented an oscillatory pattern with an amplitude of 20 kPa centered at 660 kPa, and a frequency of 18 Hz marked in red. Different factors were found to influence the pressure gauge. Among other factors, the effect of the pulses of the nozzles at 10 Hz on the the pressure gauge oscillations could be identified. Other factors influencing the pressure gauge recording could be related to the movement of

the three-piston pump, the rotational speed of the actuating PTO, or the hydraulic design of the spraying system.

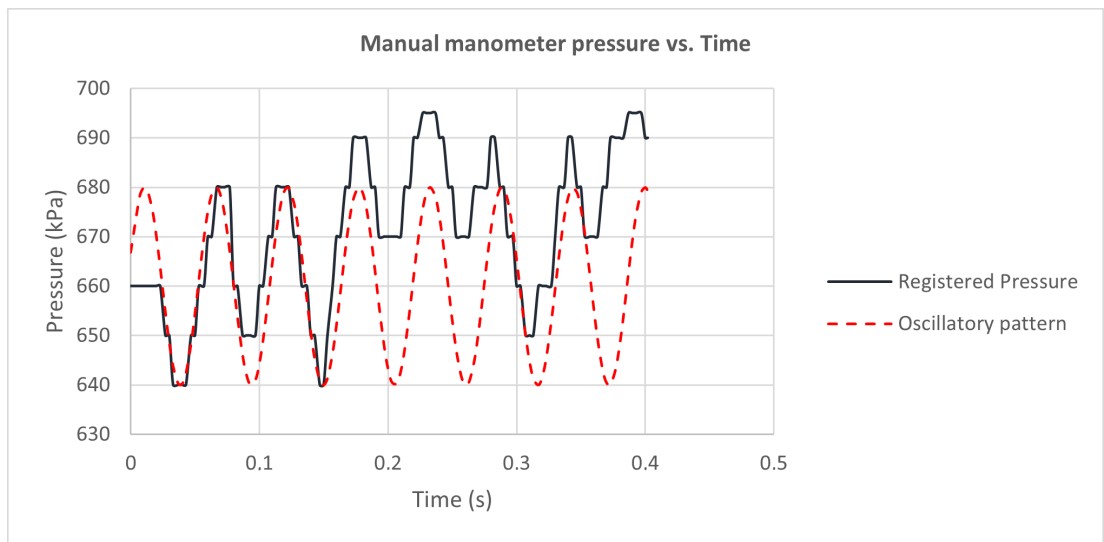

**Figure 4.** Pressure (kPa) from the manual pressure gauge in black color and oscillatory pattern registered in red color.

In addition to the video recording of the manual pressure gauge, pressure data were also logged from a digital pressure sensor. The results from the sensors confirmed those obtained with the conventional manual pressure gauge; the digital pressure gauge also registered an oscillatory pattern, as shown in Figure 5. These results are in line with previous authors (such as [17,18]) who showed the importance of pressure fluctuations in PWM-actuated spraying systems. Over the experiments, the theoretical flow rate through the nozzles ranged between a 0.12 L·min$^{-1}$ average value with a standard deviation of 0.0014 L·min$^{-1}$ for the 20% duty cycle and a 0.42 L·min$^{-1}$ average value with a standard deviation of 0.0048 L·min$^{-1}$ for the 70% duty cycle.

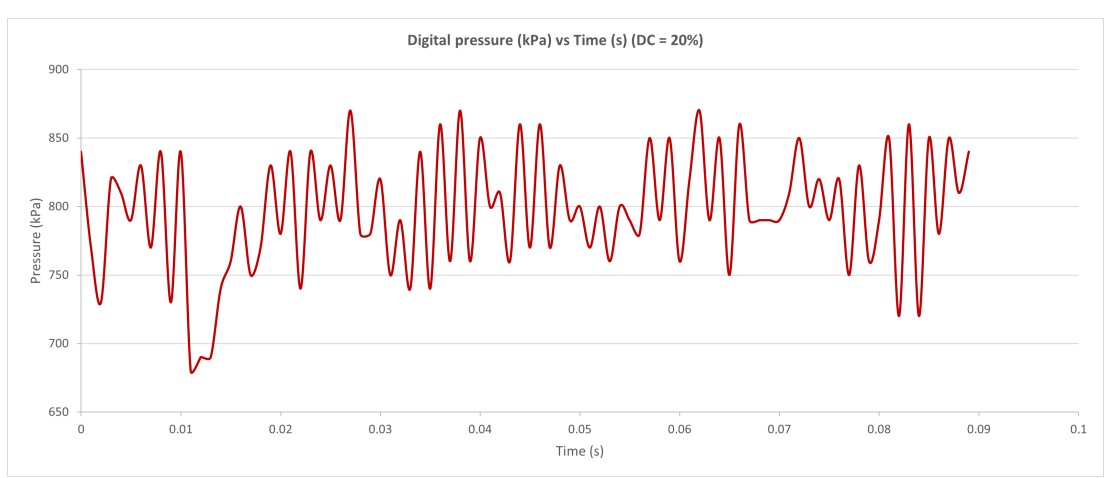

**Figure 5.** Pressure registered with time from the digital pressure sensor (KPa).

### 3.3. Vibration Analysis

In order to assess the impact of pulses on the hydraulic circuit, an accelerometer was placed at the neighborhood of a nozzle in one of the sectors (S$_1$, Figure 2). The accelerations registered in the *X* and *Y* axes (vibration plane) were very similar, but the vibration in the *Z* axis was negligible. The vibration parameters peak to peak and weighted RMS were calculated after applying a low pass filter at 100 Hz and performing signal

averaging. Significant differences were found in the acceleration when spraying and when the equipment was not spraying (Figure 6, *p*-value < 0.05). A significant increase in acceleration due to the pulsating effect of the spraying system was revealed. This result is in line with [20,21], who demonstrated how the sensor's location might register different measurements of transient pressure characteristics.

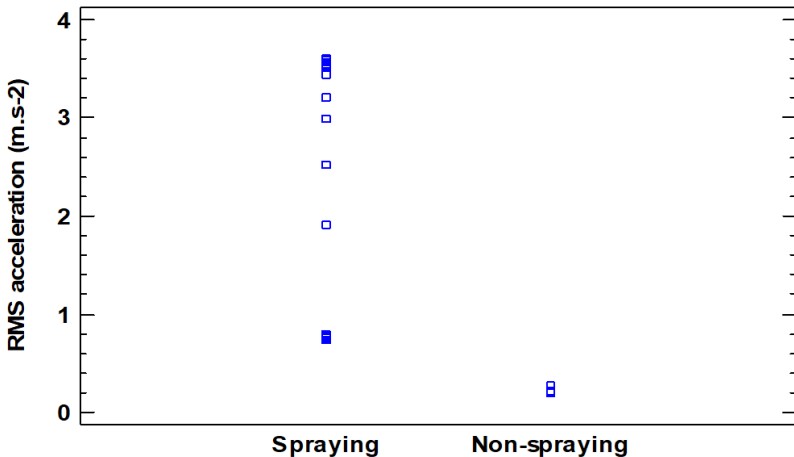

**Figure 6.** RMS acceleration (*X* axis) registered in the accelerometer located close to the nozzle when the equipment was spraying (working) and not spraying (non-spraying), all DCs included.

*3.4. Waveform Study*

The waveform analysis for the accelerometer signals resulted in a pulsating effect with a vibration cycle repeated every 100 ms, equivalent to a frequency of 10 Hz, which the pulsating frequency of the solenoid valves of the nozzles, as depicted in Figure 7, which reproduces the vibratory effect of the pulsating nozzles. The valve's opening and closing points were identified, and the opening damping, closing damping, and transient period times were calculated. A detailed analysis of the wave shows an initial impact followed by the corresponding vibration damping at the opening of the valve, which is then followed by a second lower impact related to the closing of the valve. This cycle was repeated every 100 ms, corresponding, as expected, to an activation frequency of 10 Hz.

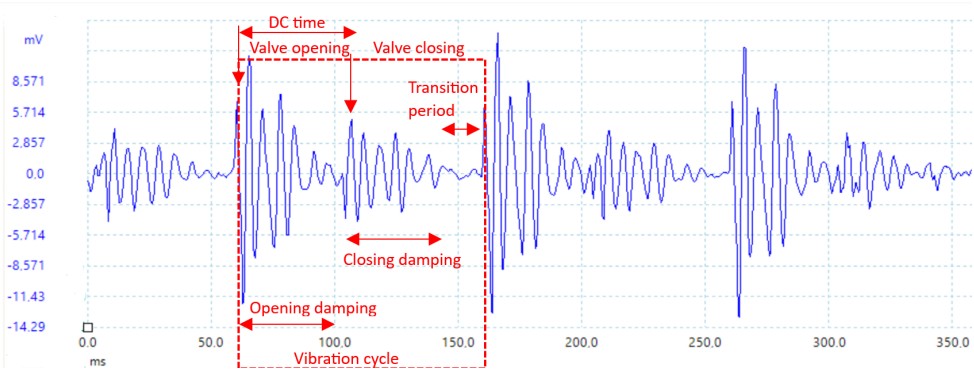

**Figure 7.** Morphology of the vibrating cycles registered with the accelerometer located close to the nozzle, 20 percent DC.

A fast Fourier transform was carried out to analyze the acceleration signal of the vibration in the frequency domain, showing a first frequency of 10 Hz and other frequencies related to the sprayer pump vibration, the constant engine idle speed, and the natural resonance frequency of the rubber pipe material's second frequency (Figure 8).

The analysis of the accelerometer signals allowed the calculation of the duty cycles at ten different vibration times per duty cycle for the three duty cycles. The duty cycle based on the accelerometer signal was calculated as the time lapse between the valve opening and closing related to the cycle time, as shown in Equation (1):

$$DC_{accelerometer}(\%) = \frac{time_{valve-closing} - time_{valve-opening}}{time_{cycle}} \times 100 \qquad (1)$$

where:

- $DC_{accelerometer}$ is the duty cycle calculated based on the accelerometer signal;
- $time_{valve-closing}$ is the time when the valve is closing (ms);
- $time_{valve-opening}$ is the time when the valve is opening (ms);
- $time_{cycle}$ is the total cycle time, repeated every 100 ms (ms).

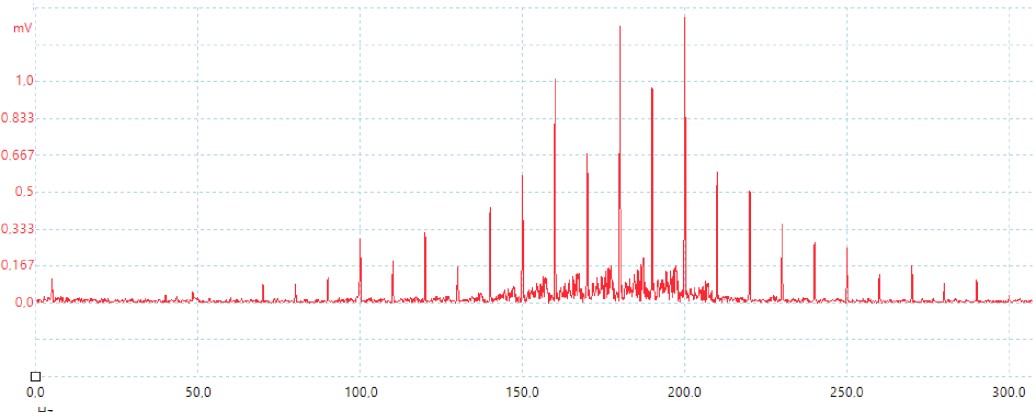

**Figure 8.** Acceleration signal of the vibration (located in the pipe close to the nozzle) in the frequency domain (fast Fourier transform developed, lineal scale, Blackman window function, 4096 spectrum bins).

Figure 9 shows the accelerations registered by the triaxial accelerometer in the three axes according to the duty cycle, in blue and red in the vibration plane (X and Y axis, respectively) and in green (Z axis). The duty cycle calculation procedure based on the accelerometer signal analysis, according to Figure 7 and Equation (1), was carried out for the tested repetitions and duty cycles.

An ANOVA was carried out to study the effect of the duty cycle on the acceleration variables (Table 1). The significant effect of the duty cycle on the peak to peak acceleration is confirmed. The amplitude of the vibration significantly decreased when the duty cycle increased. The shock derived from suddenly opening and closing the valves at high frequency is significantly higher for lower duty cycles with shorter time between opening and closing the valve.

**Table 1.** Analysis of variance of the factor DCs set up from the sprayer computer on the measured peak to peak acceleration (m · s$^{-2}$).

| Source | Sum of Squares | df | Mean Square | F-Ratio | *p*-Value |
|---|---|---|---|---|---|
| Between groups | 1025.65 | 2 | 512.83 | 1386.04 | 0.0000 |
| Intra groups | 2.22 | 6 | 0.37 | | |
| Total | 1027.87 | 8 | | | |

Based on the detailed analysis of the wave at the different duty cycles (Figures 7 and 8), the opening and closing times were identified and used to determine the duty cycles calculated using the acceleration signals, according to Equation (1). The valve opening and closing times were determined by identifying the relative maximum acceleration points

after a damping period of at least 25% of the vibration cycle repeated every 100 ms. These opening and closing points were contrasted to the recorded high-speed video images.

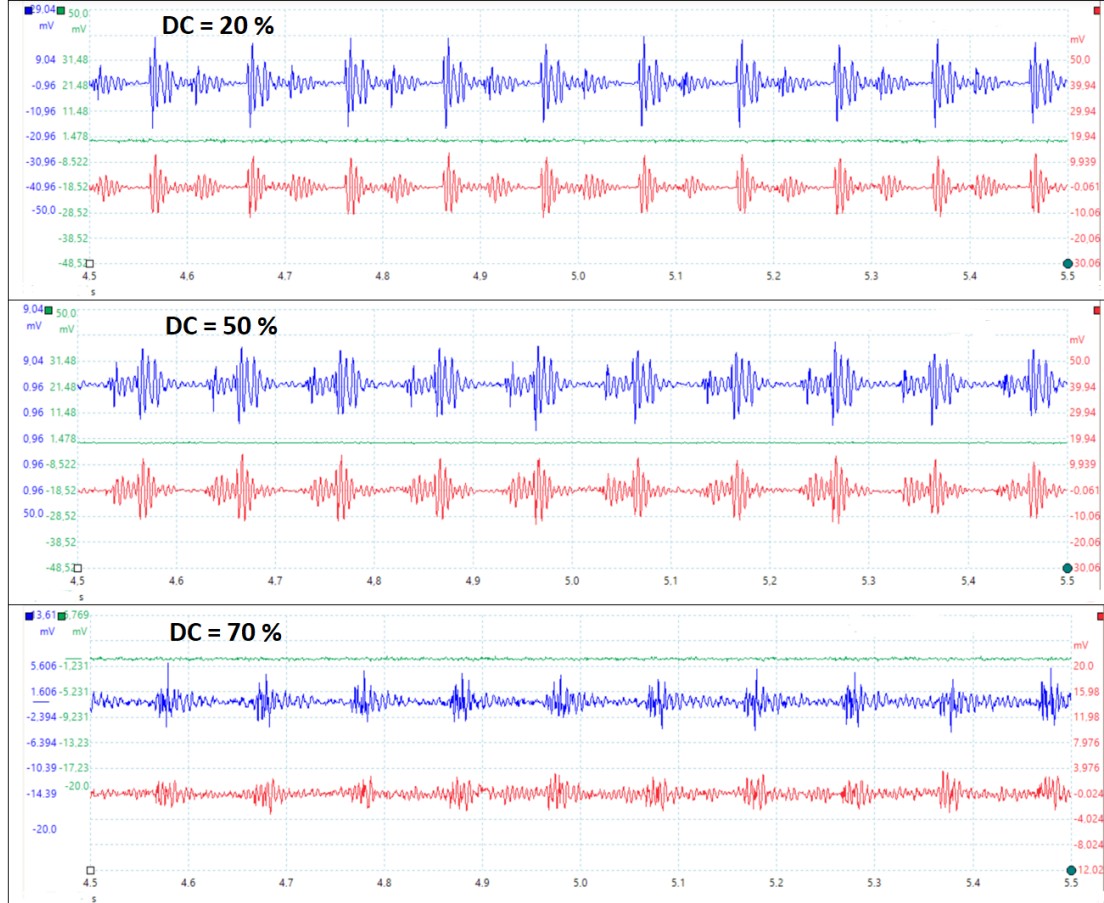

**Figure 9.** Vibration curves determined by the acceleration (mV) on the pipe for various DC.

The duty cycles calculated using the acceleration signals are shown in Table 2 and compared to the theoretical duty cycles.

**Table 2.** Theoretical duty cycle (%) and actual duty cycle (%) measured from the accelerometer signal registered between valve opening and closing.

| Theoretical DC (%) | Registered DC (%) | Maximum DC (%) | Minimum DC (%) |
|---|---|---|---|
| 20 | 43.63 | 43.29 | 43.97 |
| 50 | 70.60 | 70.26 | 70.94 |
| 70 | 89.62 | 89.28 | 89.96 |

The time lapse between valve opening and closing related to the cycle time should correspond to the theoretical duty cycle. However, this time interval between valve opening and valve closing, which was measured using the accelerometer signals, showed important differences. Significant disparities between the measured DCs calculated based on the time lapse and the theoretical percentage of the duty cycles were found for all DCs tested, as determined in Table 2. The duty cycles measured with the accelerometers were on average 21.3% higher than the theoretical duty cycles commanded from the sprayer computer. These differences between commanded and registered duty cycles were higher for lower duty cycles, with average values of 23.6%, 20.6%, and 19.6% for the 20%, 50%, and 70% theoretical duty cycles. These results confirm those found by [21,23] about discrepancies between the rated duty cycles and the actual duty cycles applied.

## 4. Discussion

An interesting finding of the vibration analysis with accelerometers is the confirmation of the divergence between commanded and applied duty cycle width due to the inaccurate opening and closing of the valves at precisely specified times. The modification of the duty cycle is caused by the vibration transient state induced by the shut-off actuation of the solenoid valves at 10 Hz. These differences between actual duty cycles registered by the accelerometers and commanded duty cycles were higher when the commanded duty cycles were lower. This fact shows the difficulty of a precise application for the crucial cases when the amount required should be lower. The modification of the duty cycle was related to the hydraulic shock derived from closing the valves suddenly at high frequency, which often resulted in the undesired effect of rising the system pressure. These results confirm those found by other authors finding differences between the rated duty cycles and the actual duty cycles applied [21–23]. The principal reason to introduce PWM solenoid valves in individual nozzles is the accurate variation of flow in real time without altering the system pressure, as a result of fast valve movements derived from high frequency actuation and the advantages of operating PWM valves from computers. However, field results showed that the addition of a complex electronic control system with PWM-operated nozzles is not enough for an optimal performance. Conventional sprayers are conceived to work in continuous flows, and so do flowmeters and pressure sensors. The introduction of high-frequency shutoff actuation at multiple points in the circuit creates vibrations and hydraulic shocks that not only complicate the accurate control of spray flows but may also damage sensors and actuators, including the solenoids themselves.

## 5. Conclusions

The vineyard blast sprayer controlled by an arrangement of 24 pulse-width-modulated nozzles divided in four sectors created a significant pulsating impact effect at nozzle level. The cycle of this impact pulse effect had the same frequency (10 Hz) as the actuating frequency of the PMW system. The major impact on the variable rating of the spray flow was determined by the morphological modification of the duty cycle, which was caused by the vibration transient state induced by the shut-off actuation of the solenoid valves at 10 Hz. This impact effect was significantly higher for low duty cycles such as 20%, and its severity was related to the hydraulic shock—or water hammer—derived from closing the valves suddenly at high frequency. The accurate measurement of the accelerometers confirmed the divergence between commanded and applied duty cycle width due to the inaccurate opening and closing of the valves at precisely specified times. Specifically, the actual duty cycles registered by the accelerometers were more than 20% higher than the theoretical ones sent by the sprayer control unit, demonstrating the offsets lapsed in the opening and closing maneuvers of the solenoids. It is crucial to consider the vibration impact effect of the pulses from the PWM nozzles. Further research should be undertaken to study technical solutions to reduce the impact effect of the opening and closing of the solenoid valves. Different pipe materials and layouts and the use of air dampers to absorb pressure fluctuations could be addressed. New hydraulic designs will likely be necessary before smart sprayers based on PWM technology may be used at their full potential.

**Author Contributions:** Conceptualization C.O. and F.R.-M.; methodology, V.S.-R. and A.T.; software, V.S.-R. and F.R.-M.; validation, C.O. and A.T.; formal analysis, C.O. and A.T.; investigation, V.S.-R., C.O. and A.T.; resources, C.O., A.T. and F.R.-M.; data curation, C.O. and A.T.; writing—original draft preparation, C.O.; writing—review and editing, C.O., A.T. and F. R-M.; visualization, A.T. and V.S.-R.; supervision, V.S.-R., A.T., C.O. and F.R.-M.; project administration, F.M-R.; funding acquisition, F.R.-M. All authors have read and agreed to the published version of the manuscript.

**Funding:** This research has been funded by the Government of Spain through the Project "Smart spraying for a sustainable vineyard and olive trees" PIVOS (PID2019-104289RB) and ADOPTA (PDC2022-133395).

**Institutional Review Board Statement:** Not applicable.

**Data Availability Statement:** The datasets generated during and/or analyzed during the current study are available from the corresponding author on reasonable request.

**Conflicts of Interest:** The authors declare no conflict of interest.

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
