# Peer review of "Vibration Analysis of Pulse-Width-Modulated Nozzles in Vineyard Blast Sprayers"

_horticulturae, doi:10.3390/horticulturae9060703_

Round 1
Reviewer 1 Report
(1)The abstract needs further refinement.
(2)Figures 4 and 6 require further modification.
(3)The discussion and conclusion should be presented in sections.
(4)Some references should be replaced.
Author Response
Responses to Reviewer 1
Thank you very much to the reviewer for their thoughtful review.
(1)The abstract needs further refinement.
- The abstract has been modified according to the reviewers´ comments.
(2)Figures 4 and 6 require further modification.
- Previous Figures 4 and 5 (new Figures 5 and 6) have been improved. Figure 5 has been corrected. Information about the Figures have been added in the text.
(3)The discussion and conclusion should be presented in sections.
- The Discussion and Conclusion are in separate sections.
(4)Some references should be replaced.
- The references have been improved and corrected according to the reviewers´ comments.

Reviewer 2 Report
The vineyard blast sprayer controlled by an arrangement of 24 pulse-width-modulated nozzles created a significant pulsating impact effect at nozzle level. The vibration characteristics of the spraying system of the vineyard sprayer are analyzed by the two indexes of acceleration signal and pressure fluctuation. The test method in the article is reasonable, the test data is detailed, and the conclusion is reliable.
but there are still related problems.
(1) In line 8-9, “evaluating the stability effect of introducing air dampers to attenuate the pressure peaks resulting from the pulses exerted by the solenoids.”
Which local data can be reflected in the following text?
(2) In line 108-110, “Vibration signals were recorded with a digital oscilloscope (PicoScope 6® systems, Picotechnology, Cambridge, United Kingdom). A high-speed digital color video cam era (CASIO EX-F1, Tokyo, Japan) recorded the spraying actuation of nozzle 3 and the oscillations of the manual pressure gauge during the tests at a rate of 300 frames/s.”
There are 24 nozzles in the designed sprayer. Does the author only select nozzle 3 for analysis ? If only nozzle 3 is selected, the representativeness of the nozzle needs to be explained.
(3) No reference to Figure 3 is found in this article. Please check whether it is a markup error .
(4) Figure 6 shows the acceleration form of a vibration period, but it does not seem to explain how to judge the opening and closing of the valve according to this form.
Author Response
Responses to Reviewer 2
Thank you very much to the reviewer for their thoughtful review. It has
lead to several substantial improvements to the manuscript.
(1) In line 8-9, “evaluating the stability effect of introducing air dampers to attenuate the pressure peaks resulting from the pulses exerted by the solenoids.”
Which local data can be reflected in the following text?
- The confused sentence has been removed from the text.
(2) In line 108-110, “Vibration signals were recorded with a digital oscilloscope (PicoScope 6® systems, Picotechnology, Cambridge, United Kingdom). A high-speed digital color video cam era (CASIO EX-F1, Tokyo, Japan) recorded the spraying actuation of nozzle 3 and the oscillations of the manual pressure gauge during the tests at a rate of 300 frames/s.”
There are 24 nozzles in the designed sprayer. Does the author only select nozzle 3 for analysis? If only nozzle 3 is selected, the representativeness of the nozzle needs to be explained.
- In order to analyze the representativeness of the vibration tested nozzle, a test was carried out to check the homogeneity among nozzles and sectors. Spray flows were measured from four nozzles in high position and four nozzles in low position. In particular, the flow was measured in nozzles 2 and 5 for all four sectors. The description of the test and the results have been detailed in new lines 112-123, 132-139.
(3) No reference to Figure 3 is found in this article. Please check whether it is a markup error.
- The reference of previous Figure 3 (new Figure 4) has been included; the error has been corrected.
(4) Figure 6 shows the acceleration form of a vibration period, but it does not seem to explain how to judge the opening and closing of the valve according to this form.
- The criteria to identify valve opening and closing in previous Figure 6 (new Figure 7) has been explained. The sentence “The valve opening and closing times were determined identifying the relative maximum acceleration points after a damping period of at least 25 \% of the vibration cycle repeated every 100 ms. These opening and closing points were contrasted to the recorded high speed video images.” has been included in the text (new lines 208-211).

Reviewer 3 Report
Review of the manuscript:
Vibration analysis of pulse-width-modulated nozzles in vineyard blast sprayers
This manuscript interested in a analyze the vibrations induced in the spraying system of a vineyard blast sprayer controlled by pulse-width-modulated nozzles, considering the instantaneous duty cycle as the control variable, and evaluating the stability effect of introducing air dampers to attenuate the pressure peaks resulting from the pulses exerted by the solenoids.
1. The findings are sufficiently novel to warrant publication.
2. The conclusions are adequately supported by the data presented.
3. The article is clearly and logically written so that it can be understood by one who is not an expert in the specific field.
4. The work provides an important contribution to its field, consistent with the scope of the journal. Discussion with different authors is needed. Errors of the model are not described. The signal analysis e. g. Fourier analysis is needed.
Comments:
Row 126: Probably Figure 3
Row 141: Figure 4, please describe the time axis by the numbers, please correct kPa on KPa.
Please explain that amplitudes in the Fig. 3 are about 2é KPa and in the Fig. 4 about 50 KPa. Please compare the frequency of the oscillations in the Fig. 3 with oscillations in the Fig 4.
Rows 169 -171: Please introduce the units of the time.
Row 173: Please realize the analysis of the accelerometer signal by some signal analysis e.g. Fourier analysis.
Row 200: Please supplement the discussion by some other authors.
Author Response
Responses to Reviewer 3
Thank you very much to the reviewer for their thoughtful review. It has
lead to several substantial improvements to the manuscript.
- The findings are sufficiently novel to warrant publication.
The conclusions are adequately supported by the data presented.- The article is clearly and logically written so that it can be understood by one who is not an expert in the specific field.
- The work provides an important contribution to its field, consistent with the scope of the journal.
Discussion with different authors is needed.
Errors of the model are not described.
The signal analysis e. g. Fourier analysis is needed.
Comments:
Row 126: Probably Figure 3
- The mistake has been corrected.
Row 141: Figure 4, please describe the time axis by the numbers, please correct kPa on KPa.
- Previous Figure (new Figure 5) has been corrected.
Please explain that amplitudes in the Fig. 3 are about 2é KPa and in the Fig. 4 about 50 KPa. Please compare the frequency of the oscillations in the Fig. 3 with oscillations in the Fig 4.
- The pressure recording was carried out to confirm the oscillatory behavior of the register. With both devices (conventional pressure gauge and digital pressure gauge), the oscillatory pattern was confirmed. Different factors are affecting the pressure oscillation (the effect of the pulses of the nozzles at 10 Hz, the movement of the 3-piston pump, the rotational speed of the actuating PTO, or the hydraulic design of the spraying system). The accuracy of the conventional pressure gauge is not comparable to the accuracy of the digital manometer. Besides, the reading of the conventional pressure gauge was carried out taking the values from the high-speed video recording of the device.
Further research should be done to analyze the pressure oscillations in the system.
Rows 169 -171: Please introduce the units of the time.
- The time units have been added (new lines 192-194).
Row 173: Please realize the analysis of the accelerometer signal by some signal analysis e.g. Fourier analysis.
- The result from the Fast Fourier Transform has been included (new Figure 8) to analyze the acceleration signal of the vibration in the frequency domain. A first frequency of 10 Hz and other frequencies (related to the sprayer pump vibration, the constant engine idle speed and the natural resonance frequency of the rubber pipe material second frequency) have been identified.
(“A Fast Fourier Transform was carried out to analyze the acceleration signal of the vibration in the frequency domain showing a first frequency of 10 Hz and other frequencies related to the sprayer pump vibration, the constant engine idle speed and the natural resonance frequency of the rubber pipe material second frequency”, new Figure 8 and new lines182-185)
Row 200: Please supplement the discussion by some other authors.
- The references have been included, new line 238.

Reviewer 4 Report
The peer revision of article:
Vibration analysis of pulse-width-modulated nozzles in vineyard blast sprayers
Brings the following conclusions:
Change decisive by necessary in line 29.
Add after hydro pneumatic, “that consists of” in line 31.
All line 39 can be removed or should be rewritten as it come after a point in line 38.
When you put a number in brackets at the beginning of a sentence you have to add the author name. 9, 11, 20, 22 and 23 are in this case.
In figure 1 please increase the size of the letters in the left drawing.
Remove a before the word signal in line 114.
Figures in lines 126 and 127 are number 3.
Sentence between lines 148-151 is not clear and too long and should be rewritten.
Change in line 152 citeref[21] by [21].
In 171 opened should replace openning.
Figure 7. Add voltage at the end of the heating.
It is not clear the difference in between groups, and intragroups. How do you form the groups?
How is it possible than the values of minimum voltage row are greater than the ones of the maximum voltage row in Table 2.
Eliminate the sentence between lines 205-207 as it comes again in the conclusion.
REFERENCES
Some references as 2, 4 and 9 are too old.
The year in line 295 is 2021.
In references 33 and 34 eliminate the letter following the year. Example 2018b
There are some errors within the text, but it is understandable. Please check my comments in the past section
Author Response
Responses to Reviewer 4
Thank you very much to the reviewer for their thoughtful review. It has
lead to several substantial improvements to the manuscript.
The peer revision of article:
Vibration analysis of pulse-width-modulated nozzles in vineyard blast sprayers
Brings the following conclusions:
Change decisive by necessary in line 29.
- According to the reviewer “decisive” has been changed to “necessary” (new line 26).
Add after hydro pneumatic, “that consists of” in line 31.
- The words “that consist of” have been added (new line 29).
All line 39 can be removed or should be rewritten as it come after a point in line 38.
- Previous line 39 (new lines 37-38) have been rewritten (“The combination of several mitigation measures and the importance of involving the farmers have been remarked by Reichenberger (2007)”.
When you put a number in brackets at the beginning of a sentence you have to add the author name. 9, 11, 20, 22 and 23 are in this case.
- The cases indicated have been corrected.
In figure 1 please increase the size of the letters in the left drawing.
- The size of the letters has been increased
Remove a before the word signal in line 114.
- The “a” before “signal” has been removed.
Figures in lines 126 and 127 are number 3.
- It has been corrected (new Figure 4)
Sentence between lines 148-151 is not clear and too long and should be rewritten.
- The sentence has been rewritten, new lines 167-169 (“Significant differences were found in the acceleration when spraying and when the equipment was not spraying, Figure 6, p-value <0.05. A significant increase in acceleration due to the pulsating effect of the spraying system was revealed”).
Change in line 152 citeref[21] by [21].
- It has been corrected.
In 171 opened should replace openning.
- It has been corrected.
Figure 7. Add voltage at the end of the heating.
- The voltage units have been added to the Figure caption.
It is not clear the difference in between groups, and intragroups. How do you form the groups?
- The different groups in Figure 9 (previous Figure 7) were adjusted according to the rated duty cycle regulated in the experiment, commanded from the sprayer computer.
How is it possible than the values of minimum voltage row are greater than the ones of the maximum voltage row in Table 2.
- Figure 9 (previous Figure 7) shows the acceleration values for different duty cycles. The acceleration values with lower duty cycles were higher than with higher duty cycles. The shock derived from suddenly opening and closing the valves at high frequency is significantly higher for lower duty cycles with shorter time between opening and closing the valve. Table 2 shows the duty cycles (measured according to equation (1)) compared to the rated duty cycles commanded from
the sprayer computer.
New information about how the accelerometer duty cycles were calculated using the acceleration signals has been added (new lines 208-212).
Eliminate the sentence between lines 205-207 as it comes again in the conclusion.
- The sentence has been removed.
REFERENCES
The year in line 295 is 2021.
- It has been corrected
In references 33 and 34 eliminate the letter following the year. Example 2018b
- The letter following the year was used to differentiate Part I and Part II.

Round 2
Reviewer 3 Report
Figure 8: Probably x-axis is in ms and no Hz, because 100ms is 10Hz
Probably identification of the frequencies and their amplitudes in the Fourier spectrum is interesting. Please describe method of computation and software.
Author Response
SPECIFIC COMMENTS
Figure 8: Probably x-axis is in ms and no Hz, because 100ms is 10Hz
- The x-axis is Hz (the graph is in the frequency domain). The first pick corresponds to the frequency of 10 Hz. The other frequencies are due to different causes (other frequencies related to the sprayer pump vibration, the constant engine idle speed and the natural resonance frequency of the rubber pipe material second frequency).
The analysis of the acceleration signal in the frequency domain was performed with a fast Fourier transformation with 401 lines in a frequency range of 0–156.2 Hz.
Probably identification of the frequencies and their amplitudes in the Fourier spectrum is interesting. Please describe method of computation and software.
- Information about the method used has been added (the software is Picoscope Oscilloscope Software 6 (Picotechnology, https://www.picotech.com/products/oscilloscope) that was already described in M6 M (used too analyze the signals from the accelerometers, acquired by the oscilloscope).
- Information about the other frequencies has been added in the manuscript.
GENERAL COMMNETS
(I) Please revise your manuscript according to the referees’ comments and
upload the revised file within 3 days.
- A document answering all the referees’ comments has been added for all the reviewers.
(II) Please use the version of your manuscript found at the above link for
your revisions.
- The indicated version has been used.
(III) Please check that all references are relevant to the contents of the
manuscript.
- The references have been checked and modified according to the reviewers’ suggestions.
(IV) Any revisions made to the manuscript should be marked up using the
“Track Changes” function if you are using MS Word/LaTeX, such that
changes can be easily viewed by the editors and reviewers.
- All the modifications in the manuscript have been marked in red color
(V) Please provide a short cover letter detailing your changes for the
editors’ and referees’ approval.
- A short cover letter was added.
